# Intranasal Delivery of Oncolytic Adenovirus XVir-N-31 via Optimized Shuttle Cells Significantly Extends Survival of Glioblastoma-Bearing Mice

**DOI:** 10.3390/cancers15204912

**Published:** 2023-10-10

**Authors:** Ali El-Ayoubi, Moritz Klawitter, Jakob Rüttinger, Giulia Wellhäusser, Per Sonne Holm, Lusine Danielyan, Ulrike Naumann

**Affiliations:** 1Molecular Neurooncology, Department of Vascular Neurology, Hertie Institute for Clinical Brain Research and Center Neurology, University of Tübingen, D-72076 Tübingen, Germany; a_ayoubi@outlook.com (A.E.-A.); moritz.klawitter@t-online.de (M.K.); jakob.ruettinger@student.uni-tuebingen.de (J.R.); wellhaueser.giulia@gmail.com (G.W.); 2Department of Urology, Klinikum Rechts der Isar, Technical University of Munich, D-81675 Munich, Germany; per-sonne.holm@i-med.ac.at; 3Department of Oral and Maxillofacial Surgery, Medical University Innsbruck, A-6020 Innsbruck, Austria; 4XVir Therapeutics GmbH, D-80331 Munich, Germany; 5Department of Clinical Pharmacology, University Hospital Tübingen, D-72076 Tübingen, Germany; lusine.danielyan@med.uni-tuebingen.de; 6Neuroscience Laboratory and Departments of Biochemistry and Clinical Pharmacology, Yerevan State Medical University, Yerevan 0025, Armenia; 7Gene and RNA Therapy Center (GRTC), Faculty of Medicine, University of Tübingen, D-72076 Tübingen, Germany

**Keywords:** glioblastoma, intranasal delivery, oncolytic adenovirus, XVir-N-31, shuttle cells

## Abstract

**Simple Summary:**

Glioblastomas (GBMs) are difficult-to-treat, deadly brain tumors and may infiltrate the whole brain. Cancer-killing (oncolytic) viruses have been used to treat GBMs. However, oncolytic virotherapy needs surgery, as the viruses have to be injected directly into the tumor. Human hepatic stellate cells were loaded with the oncolytic virus XVir-N-31 and applied into the noses of GBM-bearing mice via a non-surgical method. The virus-loaded cells rapidly migrated towards the brain tumor and invaded GBM cells located far away from the original tumor. In the brain, these shuttle cells released XVir-N-31, which then infected and killed the cancer cells. In consequence, the mice that received XVir-N-31-loaded shuttle cells via the nose showed delayed tumor growth and better survival. In addition, when the intranasal delivery was combined with an intratumoral injection of XVir-N-31, 25% of the mice did not develop any tumors and survived a long time.

**Abstract:**

A glioblastoma (GBM) is an aggressive and lethal primary brain tumor with restricted treatment options and a dismal prognosis. Oncolytic virotherapy (OVT) has developed as a promising approach for GBM treatment. However, reaching invasive GBM cells may be hindered by tumor-surrounding, non-neoplastic cells when the oncolytic virus (OV) is applied intratumorally. Using two xenograft GBM mouse models and immunofluorescence analyses, we investigated the intranasal delivery of the oncolytic adenovirus (OAV) XVir-N-31 via virus-loaded, optimized shuttle cells. Intranasal administration (INA) was selected due to its non-invasive nature and the potential to bypass the blood–brain barrier (BBB). Our findings demonstrate that the INA of XVir-N-31-loaded shuttle cells successfully delivered OAVs to the core tumor and invasive GBM cells, significantly prolonged the survival of the GBM-bearing mice, induced immunogenic cell death and finally reduced the tumor burden, all this highlighting the therapeutic potential of this innovative approach. Overall, this study provides compelling evidence for the effectiveness of the INA of XVir-N-31 via shuttle cells as a promising therapeutic strategy for GBM. The non-invasive nature of the INA of OV-loaded shuttle cells holds great promise for future clinical translation. However, further research is required to assess the efficacy of this approach to ultimately progress in human clinical trials.

## 1. Introduction

Glioblastomas (GBMs) are the most frequent malignant primary brain tumors in adults. The average survival rate of patients diagnosed with this tumor is less than 20 months, albeit with updated therapy options [1]. The infiltrative, malignant progression of this tumor and its resistance to chemotherapy and irradiation impact its devastating prognosis. Furthermore, a lack of immune surveillance by means of GBM cells enabling an immunosuppressive microenvironment is a key characteristic of GBM [2]. Additionally, a major hurdle for developing efficient anti-GBM therapies is the blood–brain barrier (BBB), which restricts the systemic delivery of many drugs to the tumor. Thus, the development of novel approaches aiming to efficiently deliver new or established therapeutics specifically to the malignant tissue are urgently needed.

A promising approach to treat GBM is oncolytic virotherapy (OVT) [3]. Either wild-type or genetically modified oncolytic viruses (OVs) are capable of replicating in neoplastic cells, ultimately spreading within the tumor and destroying it. Simultaneously, OVs leave non-neoplastic cells unharmed (for reviews, see [4,5]). Despite favorable outcomes, OVT has some limitations. Primarily, for the treatment of brain tumors, OVs must be applied intratumorally (IT) because patients often already have developed antibodies against the OVs from prior exposure that will rapidly inactivate the virus if applied intravenously [6,7]. Additionally, the entry of OVs into the brain is blocked by the BBB, which protects the brain against pathogens [6]. Moreover, OVs developed from viruses of the same origin or subtype will be rendered inactive by the patient’s immune system if applied several times [7]. Another major hurdle for OVT in GBM is the invasive and malignant fluid phenotype of the tumor [8]. Consequently, OVs applied intratumorally will not be able to reach the infiltrative GBM cells separated by non-neoplastic cells from the tumor core where the OV was administered. Therefore, it is essential to optimize OVT to try to capitalize on its full potential to treat GBM.

Oncolytic adenoviruses (OAVs), for instance, can prompt immunogenic cell death (ICD), attested to by the release of danger-associated molecular pattern (DAMP) proteins, like high-mobility group B1 (HMGB1) or heat shock proteins (HSPs) [9,10]. Subsequently, the anti-tumor immune response and anti-tumoral effects are substantially induced [11]. Furthermore, pathogen-associated molecular pattern (PAMP) molecules, like nucleic acids or viral proteins, are released by OV-infected cells, eventually stimulating the production of pro-inflammatory cytokines, such as interferons [12]. Finally, this draws dendritic cells (DCs) and advances the uptake and presentation of tumor cell debris alongside tumor-specific neo-antigens by them, ultimately priming anti-tumoral T-cell responses [13].

The intranasal administration (INA) of cells to the brain, since its groundwork discovery, has effectively proven to be non-invasive, targeted and efficacious [14,15]. INA allows a wide variety of therapeutic agents to be transported to the central nervous system (CNS), circumventing the BBB hurdle. For instance, viruses, plasmids, liposomes, cells, nanoparticles and OV-loaded cells can be delivered via INA to the CNS [15,16,17,18]. Furthermore, our earlier research confirmed the delivery of mesenchymal stem cells (MSCs) to the tumor site via INA in a GBM mouse model [19]. Accordingly, the use of shuttle cells such as MSCs to camouflage and effectively deliver OVs to the tumor has already shown promising results [20]. In our study, we aimed to maximize the potential of OVT to primarily target the invasive and infiltrative GBM cells. For this, we used the OAV XVir-N-31 (also named Ad-Delo3RGD) [21], which, in our previous work as well as in other preclinical tumor models, demonstrated extensive therapeutic efficacy, ICD induction capabilities and curative potential when applied intratumorally [10,22,23,24,25]. The deletion of the adenoviral E1A13S protein renders the replication of XVir-N-31 dependent on the nuclear YB-1 expression, which is markedly upregulated in resistant GBM cells [26]. We then utilized our optimized, highly motile, mCherry-expressing, hepatic, stellate shuttle cells, LX-2^FR^, and applied them intranasally post-XVir-N-31-infection [27]. Our newly developed LX-2^FR^ cells demonstrated, in a delayed-replication cycle, the production of infectious virus particles whilst retaining their superior migratory capabilities [27]. In the present study, we showed that a single INA of XVir-N-31-loaded LX-2^FR^ cells significantly increased the survival and reduced the tumor sizes in an orthotopic mouse model harboring GBMs derived from established LN-229 GBM cells, as well as in a more representative, highly infiltratively growing, R28 glioma stem cell (GSC)-derived GBM mouse model [28].

## 2. Materials and Methods

### 2.1. Cell Lines and Viruses

LN-229 human glioma cells (Cellosaurus ID: CVCL_0393) were a kind gift from N. Tribolet (Geneva, Switzerland) and are described in detail by Ishii et al. [29]. HEK293 cells were acquired from Microbix (Mississauga, ON, Canada; Cellosaurus ID: CVCL_0045). Both LN-229 and HEK293 cells were cultured in Dulbecco’s modified Eagle’s medium (DMEM) containing 10% fetal calf serum (FCS) and 1% penicillin–streptomycin (P/S). The R28 glioma stem cell (GSC) line was kindly provided by C. Beier (University Odense, Denmark) and maintained as tumor spheres in stem cell-permissive DMEM/F12 medium (Sigma Aldrich, Steinheim, Germany) supplemented with human recombinant epidermal growth factor (EGF) (BD Biosciences, Heidelberg, Germany), human recombinant basic fibroblast growth factor (bFGF) (R&D Systems Europe, Ltd., Minneapolis, MN, USA), human leukemia inhibitory factor (Millipore; 20 ng/mL each) and 2% B27 supplement (Thermo Fisher Scientific, Inc., Waltham, MA, USA). The R28 cell line is further described by the group of D. Beier [28]. LN-229 and R28 cells expressing the green fluorescence protein (GFP) were produced via infection with Lenti-GFP (Amsbio, Frankfurt/Main, Germany). LX-2 cells, a kind gift from Scott Friedman (the Icahn School of Medicine at Mount Sinai, NY, USA; Cellosaurus ID: CVCL_5792), were cultivated in DMEM containing 2% FCS, 1% glutamine and 1% P/S (all from Sigma Aldrich, Darmstadt, Germany) and are described in detail by Xu et al. [30]. LX-2 mCherry-positive “fast running” shuttle cells (LX-2^FR^) were generated via our previously developed and characterized method of the selection of highly migratory subpopulation of cells [19], followed by an infection with Lenti-mCherry, and are described in detail by El-Ayoubi et al. [27]. All cells were cultured at 37 °C in a humidified, 5% CO_2_-containing atmosphere. All cell lines underwent a cell line authentication analysis in May 2023 (Eurofins, Ebersberg, Germany; please refer to Appendix A) and were regularly tested for absence of mycoplasma using the MycoAlert mycoplasma detection kit (Lonza, Cologne, Germany).

XVir-N-31 was prepared, purified and titrated as previously described by Mantwill et al. and Klawitter et al. [10,25]. To load the LX-2^FR^ cells with XVir-N-31, cells were infected with a multiplicity of infection (MOI) of 200 for 5 h, which is necessary to infect > 95% of the cells but does not influence the cells’ motility [27], then were intensively washed with PBS to remove residual OVs that had not been taken up and were then directly used for INA [27].

### 2.2. Immunofluorescence and Microscopy

Mouse brains were snap-frozen on dry ice and cryosectioned (10 µm sections) using a Leica Cryomicrotome CM3050S (Leica Mikrosystems GmbH, Wetzlar, Germany). Tissue sections were washed with PBS and then blocked (3% animal serum). Immunofluorescence was performed using the following antibodies: YB-1 (59-Q; sc-101198; Santa Cruz Biotechnology, Heidelberg, Germany); Hexon (8C4; sc-51748; Santa Cruz Biotechnology, Dallas, TX, USA); HMGB1 (Invitrogen, Waltham, MA, USA, #MA5-17278); or HSP70 (Invitrogen, Waltham, MA, USA, #MA3-007). As a secondary antibody, anti-Mouse IgG Alexa Fluor^TM^ Plus 680 (Invitrogen, Waltham, MA, USA, #VC295507) was used. Double immunofluorescence staining was performed using the following antibodies: YB-1 (#NBP2-67491, Novus Biologicals, Littleton, CO, USA); Hexon (8C4; sc-51748; Santa Cruz Biotechnology, Dallas, TX, USA, #F0517); HMGB1 (Invitrogen, Waltham, MA, USA, # MA5-31967); or HSP70 (Novus Biologicals, Centennial, CO, USA, # NBP2-89951). Additionally, as a secondary antibody, anti-goat IgG Alexa Fluor^TM^ Plus 594 (Abcam, Cambridge, UK, # ab150080) was used. Tissue sections were finally mounted with mounting medium (Permount™, Thermo Scientific Fisher, Waltham, MA, USA), or nuclei were counterstained using 4’,6-Diamidino-2-phenylindol-containing mounting medium (DAPI) (Vectashield, Biozol Diagnostica GmbH, Eching, Germany). Fluorescence was analyzed using a Zeiss LSM 710 confocal microscope (Carl Zeiss AG, Oberkochen, Germany) and Zeiss Zen 3.8 software.

### 2.3. Tumor Volumetry

Mouse brains were fixed in 4% paraformaldehyde (PFA), dehydrated in 20% and 30% sucrose and cryosectioned. Tissue slices were stained with Mayer’s Hematoxylin Solution and 0.5% Eosin Y/ethanol solution (both Sigma-Aldrich, St. Louis, MO, USA) and washed under running tap water. Subsequent dehydration using an alcohol dilution series was followed by Permount mounting (Fisher Chemical; #202282). To calculate the tumor size, the starts and ends of the tumors were determined, and the area of the tumor was measured every 100 µm using ImageJ, as described by Klawitter et al. [10]. The surface area multiplied by the thickness of the section (until the next section) gave the partial volume. The sum of all the partial volumes was used to estimate the complete tumor volume.

### 2.4. Animal Experiments

Animal experiments were conducted in accordance with the German Animal Welfare Act and its guidelines (e.g., 3R principle) and were approved by the regional council of Tübingen (approval N02/20G). NOD.Cg-Prkdcscid Il2rgtm1Wjl/SzJ mice (NSG mice; the Jackson Lab, Bar Harbor, ME, USA) were bred in IVC cages in the animal facility of the Hertie Institute under pathogen-free conditions. The stereotactic implantation of GBM cells has been described in detail by Klawitter et al. and Czolk et al. [10,24]. Mice of both genders aged 2–6 months were randomized into the treatment groups. In summary, post-anesthesia and -analgesia, 1 × 10^5^ R28^GFP^ or LN-229^GFP^ cells were stereotactically implanted into the right striatum. The mice were extensively monitored to avoid and reduce pain. INA was performed as described by El-Ayoubi et al. and Yu-Taeger et al. [27,31], using either PBS, 4 × 10^6^ unloaded or XVir-N-31-loaded LX-2^FR^ shuttle cells. The intratumoral injection of XVir-N-31 (3 × 10^8^ IFU) was performed as described by Klawitter et al. [10]. Time points for INA were 28 days and, for intratumoral (IT) application, 21 days after tumor cell implantation in R28^GFP^ GBM-bearing mice, and 7 days for INA in LN-229^GFP^ GBM-bearing mice.

### 2.5. Statistical Analysis

For all in vivo experiments, the group and sample sizes are indicated in the figure legends. Kaplan–Meier survival studies were analyzed using the log-rank (Mantel–Cox) test. Further statistical analyses were conducted with a two-tailed Student’s *t*-test or one-way ANOVA using GraphPad Prism 9.5.1 (GraphPad Inc., San Diego, CA, USA). The results are represented as the mean ± standard error mean (SEM), and *p*-values of <0.05 were considered statistically significant (n.s.: not significant; * *p* < 0.05; ** *p* < 0.01; *** *p* < 0.001; **** *p* < 0.0001).

## 3. Results

### 3.1. XVir-N-31 Reaches the GBM after INA of OV-Loaded LX-2^FR^ Cells

As adenovirus subtype 5, the origin of XVir-N-31, does not efficiently replicate in mouse cells, human GBM xenograft models in mice were used in this study. As previous in vivo experiments showed that the INA of optimized, “fast running”, highly motile, hepatic, stellate shuttle cells (LX-2^FR^) reached LN-229-derived GBMs in mice, hitting both the tumor core as well as its infiltration zones [27], we wanted to investigate whether loading these cells with XVir-N-31 displayed a similar outcome, as well as whether INA can be used as a therapeutic to cargo OVs to the tumor. Therefore, we performed a single INA of 4 × 10^6^ LX-2^FR^ shuttle cells that had been infected with a 200 MOI of XVir-N-31 (LX-2/XVir) 5 h prior to the INA to LN-229^GFP^ GBM-bearing mice at a time point at which the tumor developed a size of approximately 2 mm in diameter. We performed immunofluorescence analyses to identify the shuttle cells, XVir-N-31 and its replication in the tumor region at several time points post-treatment. A strong colocalization of the XVir-N-31 and LX-2^FR^ cells in close proximity to the tumor was observed at day 3 post-INA (Figure 1). At this time point, the presence of XVir-N-31, indicated by the adenoviral hexon protein, was exclusively detected in the LX-2^FR^ cells. At later time points (days 12 and 18 post-INA), no LX-2^FR^ cells were detectable anymore, most likely due to the OV-mediated cell lysis and the release of OV progeny. At these later time points, XVir-N-31 had spread throughout the tumor, with its replication now confined to GBM cells.

Furthermore, in a satellite tumor, which is often seen in multifocal GBMs and is suggested to be derived from infiltrating GBM cells [32,33], the hexon protein, which indicates XVir-N-31 replication, was also evident 18 days after treatment (Figure 2). No hexon staining outside the tumor areas was observed (Figure 1, day 18). The prominent hexon staining within the core tumor and its satellite not only indicates the presence of XVir-N-31, but also its infectious chain reaction in GBM cells.

### 3.2. In LN-229-Derived GBMs, the INA of LX-2/XVir Induced Immunogenic Cell Death, Reduced Tumor Growth and Extended the Survival of Tumor-Bearing Mice

To investigate the therapeutic potential of the INA of LX-2/XVir, we firstly examined its capability to induce immunogenic cell death (ICD). ICD induction in tumor cells through OVs is a warrant of their therapeutic efficiency and an important driver in the induction of a specific anti-tumoral immune response [6,10]. A substantial indication of ICD is the release of DAMPs. We examined the DAMPs HMGB1 and HSP70 in addition to the immunogenic protein YB-1 [34,35]. In accordance with the spreading of XVir-N-31 in the LN-229 GBMs after the INA of LX-2/XVir (Figure 1), HMGB1 was clearly evident within the tumor area 12 days after the INA of LX-2/XVir, but not if unloaded shuttle cells were intranasally applied. HMGB1 staining clearly manifested at later time points after the INA. Interestingly, 18 days after the INA, some hot spots of the HMGB1 staining were visible, whilst in most of the tumor area, HMGB1 was uniformly distributed (Figure 3, lower two panels).

Comparable results were observed for HSP70 and YB-1. All three proteins colocalized with the adenoviral hexon protein, indicating the presence of XVir-N-31-infected cells in this area (Figure 4). Furthermore, in the infiltration zone adjacent to the tumor core, we also detected HMGB1, HSP70 and YB1, consistently colocalizing with the adenoviral hexon protein (Appendix A), indicating that ICD was induced exclusively in the XVir-N-31-infected GBM cells.

These promising results prompted us to examine the therapeutic impact of LX-2/XVir-based INA in LN-229^GFP^ GBM-bearing mice by determining the survival as well as the tumor growth. Mice that received a single INA of LX-2/XVir showed significantly smaller tumors than the sham-treated mice, in which the tumor cells spread in nearly half of the hemisphere (Figure 5A,B). Additionally, the INA of LX-2/XVir significantly extended the survival of the mice compared to those animals that received either PBS (sham) or unloaded shuttle cells (Figure 5C,D). Moreover, the treatment was not harmful to the animals, as none of the mice showed any treatment-related symptoms in behavior or even weight loss (Figure 5E).

### 3.3. INA of LX-2/XVir Provides Therapeutical Impact in Mice Bearing Highly Infiltrating, Glioma Stem Cell-Derived GBMs

LN-229 GBMs present tumors that show tumor cell infiltration in mice and can be used as a tumor model for rapidly growing tumors. Unfortunately, LN-229^GFP^-derived GBMs do not show the high level of infiltrative growth that is observed in most human patients. In contrast, glioma stem cells (GSCs), like the R28 cells that were derived from primary human glioblastomas and cultured as neurospheres in a stem cell medium, more closely mirror the phenotype and genotype of primary tumors than do established GBM cell lines that were cultured in a high-glucose and FCS-containing medium. It has been described that fetal calf serum induces the differentiation of stem cells and therefore influences the phenotype of serum-cultured cells [36]. In the brains of immunocompromised mice, GSCs developed tumors with an elevated infiltrative growth capacity compared to LN-229-derived GBMs [36,37]. Therefore, GSC-derived GBMs might present a more clinically relevant model of experimental GBM. As our goal was to reach, via the INA of LX-2/XVir, tumor cells not only in the original tumor area, but also the infiltrated tumor cells not eradicated via intratumorally injected OVs, we used NSG mice bearing R28^GFP^ GSC-derived tumors, which grow slowly [25] but highly infiltrate the surrounding brain parenchyma and are even able to invade the contralateral hemisphere (Appendix A). Even if the volumes of the R28^GFP^ tumors in the sham-treated animals were highly variable, the tumor sizes were significantly reduced after the INA of LX-2/XVir (Figure 6A). At the time point at which we measured the tumor volumes, which was the day that the first mouse displayed tumor burden symptoms, some tumors were tiny and nearly not detectable (Figure 6B).

As we successfully used intratumoral (IT) injections of XVir-N-31 to treat R28 GBMs [25], we were interested in whether the therapeutic impact of LX-2/XVir-based INA is comparable to that of intratumorally (IT) applied XVir-N-31, and whether a combination of INA and IT might further enhance the therapeutic impact of this OVT. As shown in Figure 6C, both the LX-2/XVir-based INA and the intratumoral injection of XVir-N-31 prolonged the survival of the R28^GFP^ GBM-bearing mice. The median survival of the sham-treated mice that received only the vehicle (PBS) was 151 days, the median survival for the mice that intratumorally received XVir-N-31 was 214 days and the median survival for the group of animals we treated via LX-2/XVir-based INA was 224 days, signifying the superior effect of the INA:LX-2/XVir-based OVT (Figure 6D). Again, the mice did not suffer from the therapy, and they did not lose weight (Figure 6E). Notably, the combination of IT and INA further prolonged the animals’ survival, and in 2/8 mice at the end of the experiment (day 350 post-GBM-cell-implantation), no tumors were detected (Appendix A).

### 3.4. XVir-N-31 Is Present Long Term after INA of LX-2/XVir

Finally, as the INA of LX-2/XVir gave a significant survival advantage, and as we observed no tumors in 25% of the mice in the combination group, we investigated whether XVir-N-31 is still actively present for a long time after treatment. Therefore, hexon staining was performed in the tumor areas of the R28^GFP^ GBM-bearing mice that had to be sacrificed after the display of tumor symptoms. Surprisingly, with no trace of the LX-2^FR^ shuttle cells, the hexon staining was specifically colocalized with the R28^GFP^ cells, independent of whether the mice received the OV intratumorally or via INA (Figure 7). However, we identified slightly more hexon staining in the INA and the combination of INA and IT than in the single-IT-treated animals. In an animal of the INA group, we also observed hexon staining in an infiltration area of an R28^GFP^ GBM (Figure 7B).

We also examined whether the capability of XVir-N-31 to induce ICD was maintained at this late time point after treatment. Indeed, in the mice that received XVir-N-31 intratumorally, HMGB1, HSP70 and YB-1 were evident in the core tumor for a long time post-treatment. Similarly, the INA:LX-2/XVir (140 d post-treatment)- and INA/IT combination (182 d post-treatment)-treated mice also showed HMGB1, HSP70 and YB-1 staining in the tumor, which, in most tumor areas, was more or less equally distributed, whereas in a few areas, like in LN-229^GFP^ tumors at day 18 (Figure 3), hot spots of HMGB1 staining were visible (Figure 8, Appendix A). Interestingly, whilst R28^GFP^ GSCs spread from the core tumor throughout the brain, demonstrating the strong infiltrative ability of these cells, neither HMGB1, HSP70, YB-1 nor hexon stains were uniformly evident throughout the brain or the complete tumor. Even in animals that received the combination treatment, there were tumor areas in which none of these proteins was detected (Appendix A). Collectively, these observations indicate that after more than 6 months post-INA-based-OVT, XVir-N-31 is able to replicate and additionally induces ICD in infected tumor cells.

## 4. Discussion

Even with recent advances, and in spite of promising clinical and preclinical data [38,39,40], OVT has some limitations in the treatment of GBM. Firstly, most patients possess antibodies against the therapeutically available OVs, especially against adenoviruses, leading to their fast inactivation. Therefore, OVs targeting solid tumors cannot be applied systemically but rather have to be injected intratumorally. However, when applied intratumorally, non-neoplastic cells surrounding the injection site or the tumor core hamper the spreading of OVs to invasively growing GBM cells that are often located distantly from the intratumoral virus injection site. Yet, these invasive GBM cells frequently harbor stem cell characteristics and are mainly therapy-resistant [36,37]. Therefore, it is crucial to eliminate the invaded GBM cells to provide a “longer term” survival benefit to the patient. Secondly, an intact BBB, which is present in the tumor infiltration zone and healthy brain where invaded GBM cells are localized, restricts the delivery of therapeutics, including OVs. The intact BBB is an additional hurdle for an effective GBM cell-targeted therapy. In our recent study, we investigated the potential of the intranasal delivery of optimized shuttle cells loaded with the OAV XVir-N-31 to not only reach GBM cells in the tumor core, which are hit by the intratumoral delivery of OVs, but also to reach and ultimately destroy invaded GBM cells distant from the original core tumor. In several foregoing studies, it has been shown that XVir-N-31 competently replicates in glioma cells, eventually inducing oncolysis and prolonging the survival of GBM-bearing mice [10,24,25]. Compared to the intratumoral injection or convection-enhanced delivery of OVs to brain tumors that need surgery [41], INA is non-invasive and a more targeted method to transport drugs like OVs into the brain than their systemic delivery.

To confirm that the INA of LX-2/XVir is an effective method to deliver XVir-N-31 to (invaded) GBM cells, that it replicates in and eliminates GBM cells and that it might provide a survival benefit that is equal to or even better than its intratumoral delivery counterpart, we used two orthotopic xenograft GBM mouse models. Whilst LN-229^GFP^ tumors show rapid growth and minor invasive potential [42], R28^GFP^ GSC-derived tumors grow slowly but invasively [25] (Appendix A). Three days after the INA, the XVir-N-31-loaded shuttle cells were present in close proximity to the tumor area (Figure 1), indicating sufficient transport into the brain and towards the tumor. A time period of 72–96 h has been shown to cover one replication cycle of XVir-N-31 in LX-2 cells and defines the time point at which the cells will stop migration, as they will be lysed via OV replication and virus release. This might cast doubt on whether, in humans, this period is long enough for the shuttle cells to travel from the nose to the brain and towards invaded GBM cells. However, LX-2^FR^ shuttle cells delivered via INA were observed in all the brain areas of mice at about 6 h after INA, with no effect on the cells’ migratory capacity after loading them with XVir-N-31 [27]. We believe that the observed rapid transfer into the brain might give OV-loaded LX-2^FR^ shuttle cells enough time, even in humans, to travel from the nose to the brain and, once there, reach GBM cells. Shortly after, XVir-N-31 spread to the GBM cells in the tumor core, the infiltration zones as well as the microsatellite tumors, where it further replicated, as shown by the hexon staining (Figure 1, Figure 2 and Figure 4, Appendix A). In the R28^GFP^ tumors, XVir-N-31 replication was detectable even a long time (up to more than 180 d) post-treatment, indicating a sufficient transport of XVir-N-31 and its infection of GBM cells via the INA of OV-loaded optimized shuttle cells, as well as a sufficient OV amount to start the chain reaction of virus replication in tumor cells. The successful cargo of XVir-N-31 to GBM cells was confirmed by the prolonged survival of both the LN-229^GFP^ and R28^GFP^ GBM-bearing mice that received the INA of LX-2/XVir. Additionally, these mice harbored smaller tumors than the mice that intranasally received either PBS or unloaded shuttle cells (Figure 5 and Figure 6).

We were interested in determining whether INA, as a non-invasive method for delivering cargo OVs into the brain, is equally effective in its therapeutic impact compared to the intratumoral delivery of OVs. To investigate this, we used the highly invasive R28^GFP^ GBM mouse model, from which we knew from preliminary studies that a single intratumoral injection of XVir-N-31 significantly prolonged survival [25], and we applied the OAV either via the INA of LX-2/XVir or via an IT injection of XVir-N-31. In addition, we combined both application methods. For the combination, we first performed the IT injection of XVir-N-31, and seven days later we performed the INA of LX-2/XVir. As shown in Figure 6, the therapeutic effects of the INA and IT delivery of XVir-N-31 were comparable based on the survival analysis, suggesting that the INA-of-LX-2/XVir approach is a feasible option to treat GBMs. Notably, the combination of the INA/IT-based delivery of XVir-N-31 further extended the mice’s survival, and in 2 out of 8 mice, no tumors were detectable one year after therapy. However, in those mice of the combination group that developed tumors and had to be sacrificed, areas in the tumor presented no virus detection, indicating that, in this highly infiltrative tumor, further hurdles, like the partial encapsulation of GBM cells by non-neoplastic cells, or the very rapid growth of the GBM cells in some regions, might limit the therapeutic impact. Therefore, in future studies, the IT/INA combination should be further optimized, for example, by using the recurrent INA of LX-2/XVir or shuttle cell loading with different OV species. The former bears the putative potential that, even if XVir-N-31 will be applied to the immunoprivileged brain, and the OAV is disguised by the INA of virus-loaded shuttle cells, neutralizing antibodies might be generated after the first cycle of OVT.

As the therapeutic impact of an OV-based therapy is not only evoked by the oncolysis-mediated killing of tumor cells, but also, and probably largely, by the induction of ICD and the subsequent enabling of an anti-tumoral immune response [10,43,44], we also determined whether the XVir-N-31-based OVT induces ICD and the duration after treatment that the induction of the ICD lasts. Indeed, ICD, identified via the detection of DAMPs such as HMGB1 and HSP70, or via the immunogenic protein YB-1 [9,12,45], was induced in both the core as well the infiltration zones of the LN-229^GFP^ and R28^GFP^ GBMs (Figure 3, Figure 4 and Figure 8, Appendix A), where it colocalized with the adenoviral hexon protein, indicating that ICD is exclusively induced in XVir-N-31-infected cells. In the GBM-bearing mice that received the LX-2/XVir-based OVT, the massive induction of ICD, as indicated by HMGB1, was observed at later time points after treatment. Interestingly, in some tumor areas, but independently of whether INA or IT was performed, hot spots of HMGB1 staining were observed (Figure 3, second-last panel). As ICD induction is strictly dependent on XVir-N-31 replication [10], we believe that these hot spots indicate ongoing OV replication in these areas. In addition, the detection of HMGB1 and HSP70 months after treatment that we observed in the R28^GFP^ GBM-bearing mice that received XVir-N-31-based OVT suggests that ICD induction lasts as long as virus replication occurs (Figure 8, Appendix A). Regrettably, there were also tumor areas where no virus replication was detectable and therefore no ICD induction was visible (Appendix A), indicating that even using INA-based OVT or a combination of INA and IT induces a uniform distribution of the OV in the tumor area.

Unfortunately, in our xenograft models using immunocompromised mice, it was not possible to measure the immunostimulating effects of XVir-N-31 on the tumor growth and survival. Nevertheless, the therapeutic benefit of ICD induction via XVir-N-31 was demonstrated in our previous study, in which we used immunohumanized mice and conducted an XVir-N-31-based OVT [10]. We believe that in an immunocompetent system, particularly the IT:XVir plus INA:LX-2/XVir combination might further enhance the therapeutic impact we still observed in immunocompromised mice (Figure 5 and Figure 6, Appendix A).

Recently, INA succeeded in an FIH clinical trial and was proven safe in pediatric patients after the INA of allogeneic stem cells [46]. To this end, our study provides the first direct evidence and the translational background for the establishment of intranasal, OV-loaded, somatic differentiated cells (such as LX-2FR) for targeting invaded GBM cells alone or in combination with the local OV therapy of the tumor core.

## 5. Conclusions

In conclusion, the INA of LX-2/XVir represents a promising therapeutic approach for GBM. This study demonstrated that this non-invasive, safe and effective delivery method significantly extends the survival of GBM-bearing mice, offering new hope in the fight against this aggressive brain tumor. The utilization of XVir-N-31 harnesses the tumor-selective replication and subsequent destruction of GBM cells, while the optimized LX-2^FR^ shuttle cells efficiently deliver XVir-N-31 into the tumor and also invade GBM cells, which are often located at a far distance from the original tumor. This non-invasive, intranasal route not only enhances drug delivery but also minimizes systemic side effects and provides the potential for repeated treatments. The findings accentuate the potential clinical significance of INA in human patients, providing a less invasive and safer approach than conventional treatments. Further translational research is warranted to validate these results in human clinical trials, but this study represents a crucial step towards a breakthrough therapy for GBM patients, potentially improving their quality of life and survival rates.

## Figures and Tables

**Figure 1 cancers-15-04912-f001:**
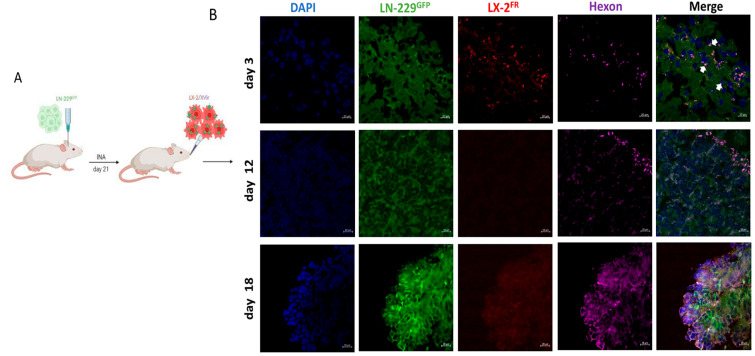
In LN-229^GFP^ GBM-bearing mice, XVir-N-31-loaded LX-2^FR^ shuttle cells reach the tumor and the OV spreads to tumor cells. (**A**) Schematic timeline of the treatment. (**B**) Detection of shuttle cells (LX-2^FR^, red) and XVir-N-31 (hexon, magenta) in the tumor area 3, 12 and 18 days post-INA of 4 × 10^6^ LX-2^FR^ cells loaded with 200 MOI XVir-N-31. Arrows indicate colocalization of XVir-N-31 and shuttle cells (n = 3 mice per group; representative pictures are shown; bars = 20 µm).

**Figure 2 cancers-15-04912-f002:**
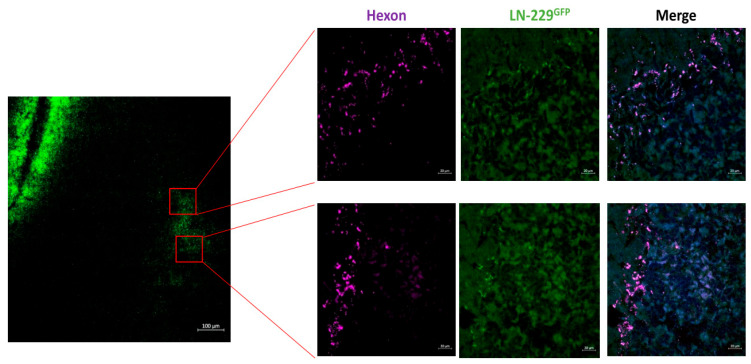
After INA of LX-2/XVir, the OV also replicated in a microsatellite tumor adjacent to the implanted tumor. Detection of XVir-N-31 via hexon staining in an LN-229^GFP^-derived microsatellite tumor 18 days post-INA (overview: bar = 100 µm; magnifications: bars = 20 µm).

**Figure 3 cancers-15-04912-f003:**
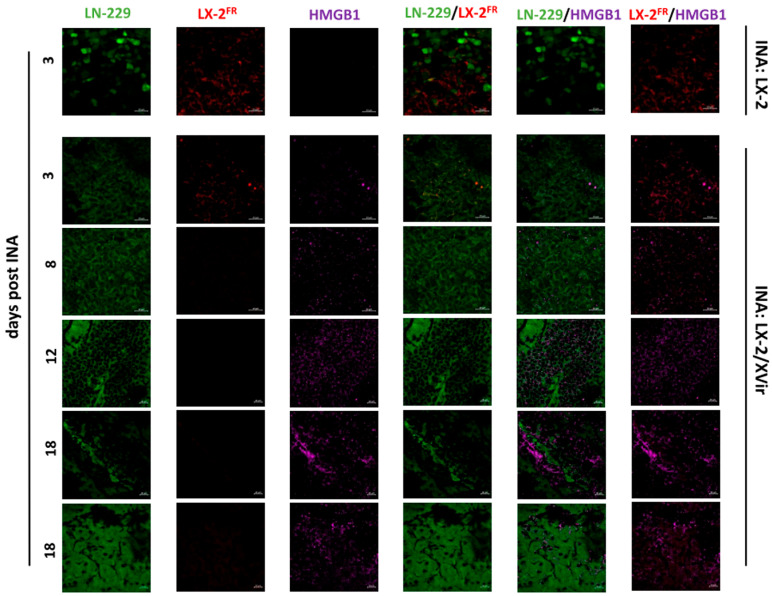
HMGB1 is present in LN-229^GFP^ GBMs post-INA of LX-2/XVir. Upper panel presents immunofluorescence (IF) stains of mice that received INA of unloaded shuttle cells, whereas lower panels present IF analyses of mice that received INA of LX-2/XVir. IF analyses were performed at the indicated time points after INA (n = 3 mice per group; representative pictures are shown; bars = 20 µm).

**Figure 4 cancers-15-04912-f004:**
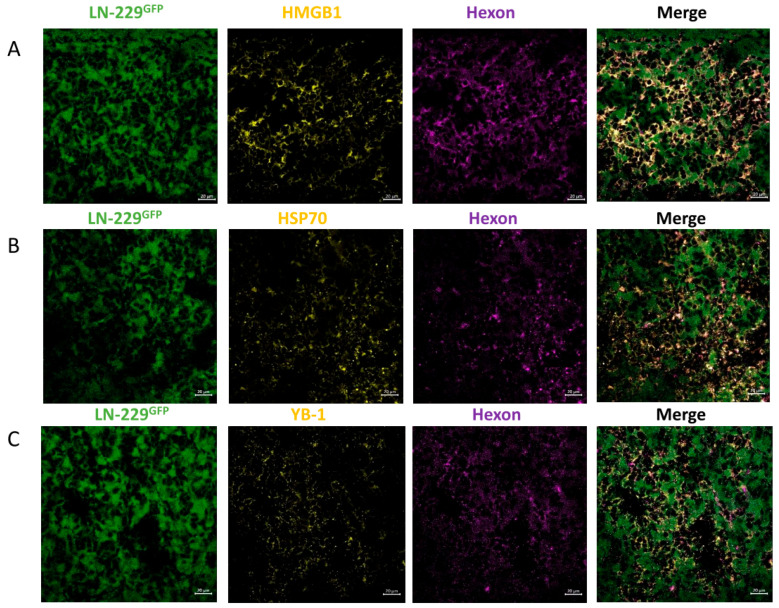
Detection of HMGB1, HSP70 and YB-1 in the tumor area of LN-229^GFP^ GBMs. (**A**) HMGB1 plus hexon, (**B**) HSP70 plus hexon and (**C**) YB-1 plus hexon stains in LN-229^GFP^ GBMs 18 days post-INA of LX-2/XVir (n = 3 mice per group; representative pictures are shown; bars = 20 µm).

**Figure 5 cancers-15-04912-f005:**
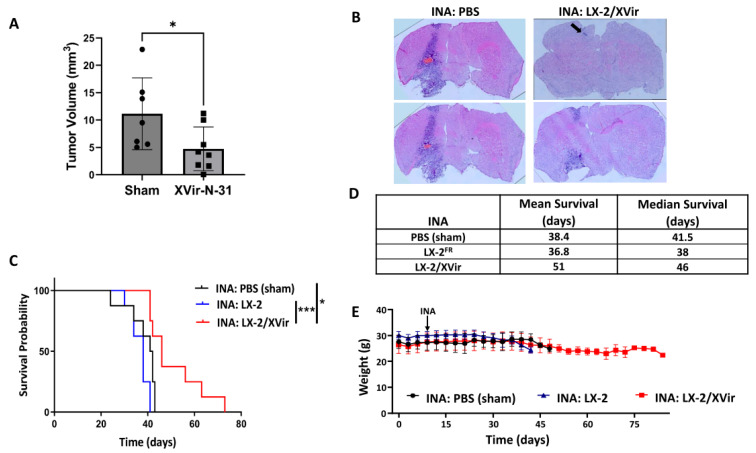
INA of LX-2/XVir significantly prolongs survival and reduces tumor volume in LN-229^GFP^ GBM-bearing mice. (**A**) INA of LX-2/XVir significantly reduced the tumor volume compared to sham-treated mice. All mice were sacrificed at the time point at which the first mouse developed tumor-associated symptoms. Brains were stained with HE and tumor volumetry was performed as indicated in the Materials and Methods section (n = 7–8 mice per group, means and SEMs, *t*-test, * *p* < 0.05). (**B**) Representative images of the tumors. The arrow indicates a small tumor we detected in one animal of the treatment group. Pictures were taken at 10 x magnification and were combined using a tile scan function. (**C**) Kaplan–Meier survival analysis of mice bearing LN-229^GFP^ GBMs that received either INA of PBS (sham), INA of 4 × 10^6^ unloaded LX-2^FR^ shuttle cells or INA of 4 × 10^6^ LX-2^FR^ cells infected with 200 MOI of XVir-N-31 (n = 8 mice per group, ANOVA; * *p* < 0.05; *** *p* < 0.001). (**D**) Mean and median survival. (**E**) Weight curves of the mice of the experimental groups (n = 7–8 mice per group, means and SEMs).

**Figure 6 cancers-15-04912-f006:**
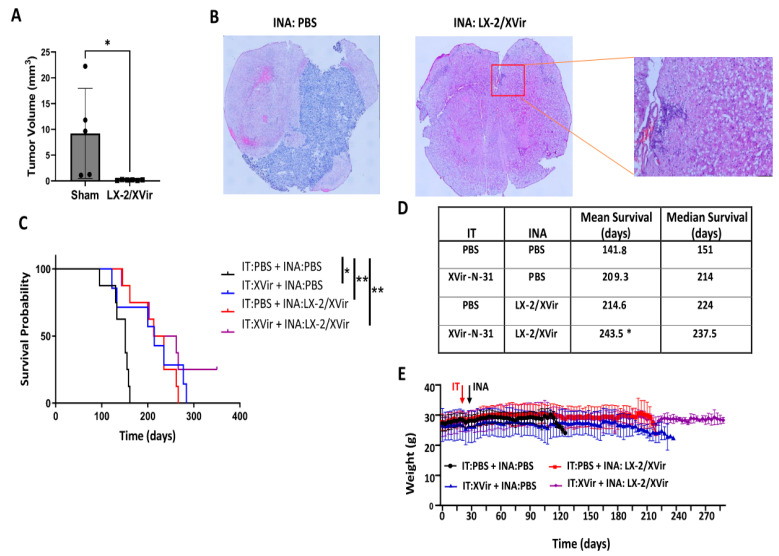
INA of LX-2/XVir significantly prolongs survival and reduces tumor volume in mice bearing R28^GFP^-derived GBMs. (**A**) Tumor volumetry showed significantly smaller tumors in mice that received INA of LX-2/XVir compared to sham-treated mice that received INA of PBS. Mice were sacrificed at the time point at which the first mouse presented tumor burden symptoms (n = 5–6 mice per group; means and SEMs; ANOVA; * *p* < 0.05). (**B**) Representative HE images of the tumors. (**C**) Kaplan–Meier survival analysis of mice bearing R28^GFP^ GBMs that received either INA of PBS plus a single IT injection of PBS (sham), INA of LX-2/XVir plus an IT injection of PBS, INA of PBS plus an IT injection of 3 × 10^8^ IFU XVir-N-31 (IT:XVir) or the combination of INA of LX-2/XVir plus IT of XVir (n = 7–8 mice per group; ANOVA; ** *p* < 0.01). (**D**) Mean and median survival. The mean for the combination group was calculated presuming the experiment had to be terminated at 350 days post-tumor-implantation (*: Mean survival calculation at experiment termination at day 350 post tumor-implantation. (**E**) Weight curves of the mice of the experimental groups (n = 7–8 mice per group, means and SEMs).

**Figure 7 cancers-15-04912-f007:**
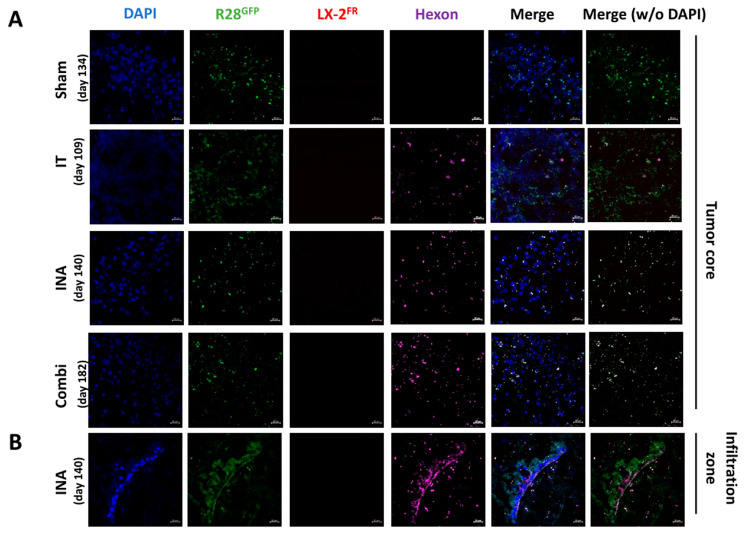
XVir-N-31 still replicates in R28^GFP^-derived GBMs a long time after OVT. The mice were treated as indicated in Figure 6. (sham: INA:PBS + IT:PBS; IT: INA:PBS + IT:XVir; INA: INA:LX-2/XVir + IT:PBS; Combi: INA:LX-2/XVir + IT:XVir). Days indicate the time points at which the animals had to be sacrificed due to tumor-associated symptoms, and, by this time, the immunofluorescence analysis had been performed. (**A**) Absence of hexon staining in the tumor region of sham-treated mice, whilst hexon staining was detectable in all OVT-treated animals. (**B**) Hexon staining in the infiltration zone of an R28^GFP^-derived GBM 140 days after INA:LX-2/XVir plus IT:PBS (n = 3 mice per group; representative pictures are shown; bars = 20 µm).

**Figure 8 cancers-15-04912-f008:**
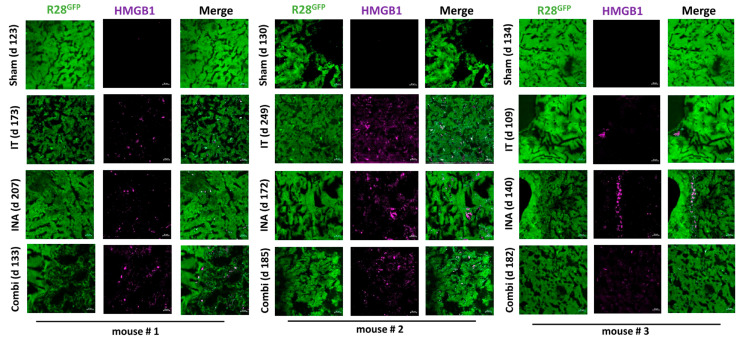
XVir-N-31 still induces ICD in R28^GFP^-derived GBMs a long time after OVT. HMGB1 is detected in 3 different mice of each group harboring R28^GFP^ tumors a long time after treatment. Absence of HMGB1 in sham (INA:PBS + IT:PBS)-treated mice, but detection of HMGB1 in all treatment groups for a long time post-XVir-N-31-administration, regardless of the administration method, indicates the induction of ICD in XVir-N-31-infected GBM cells. The time points indicate the period post-treatment that the mice had to be sacrificed due to tumor-associated symptoms (n = 3 mice per group; representative pictures are shown; bars = 20 µm).

## Data Availability

The datasets generated and/or analyzed during the current study are available in the Mendeley data repository (Mendeley Data, V1, doi:10.17632/4244khm2h2.1).

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
