# Peer review of "Intranasal Delivery of Oncolytic Adenovirus XVir-N-31 via Optimized Shuttle Cells Significantly Extends Survival of Glioblastoma-Bearing Mice"

_cancers, 2023, doi:10.3390/cancers15204912_

Round 1

Reviewer 1 Report

Oncolytic viruses (OVs) have been used to treat GBMs by intratumoral injection, because the entry of OVs into the brain is blocked by the BBB that protects the brain against pathogens. This study used human cells that loaded with the oncolytic virus XVir-N-31. Virus-loaded cells were applied into the noses of GBM-mice by a non-surgical method, which rapidly transported towards the brain tumor and also to invaded GBM cells. In the brain these shuttle cells released XVir-N-31 which then infects and kills the cancer cells. In consequence, mice that received XVir-N-31 loaded shuttle cells via the nose showed a delayed tumor growth and better survival. However, there are several major defects that the authors should be addressed.

1.       To load the LX-2FR cells with XVir-N-31, cells were infected with a multiplicity of infection (MOI) of 200 for 5 h, then directly used for INA. The infection efficiency of the viruses has not been identified.

2.       In Figure 1B, the consistency of the most obvious positive bright spots in day3 among LN229GFP, LX-2FR, and Hexon have a great doubt. Please provide more evidence. In addition, the histopathological structure of the LN229GFP transplanted tumors should be given HE staining images for further demonstration.

3.       In Figure 3, the second row of images, the expression of LX-2FR and HMGB1 showed a high consistency. Since LX-2FR is human cell, while HMGB1 marks a murine-derived protein, it is impossible to be in exactly the same position, so the images are not trustworthy. Similarly, in Figure 4, HMGB1 and HSP70 are labeled proteins expressed by mouse cells, while Hexon is a protein expressed by viruses. Since viruses are replicative in tumors and can spread diffusively, the viruses have a wider distribution range, and the Hexon expressed by the viruses cannot be distributed in the same position as the protein expressed by mouse tissue cells.

4.       As authors said, the entry of OVs into the brain is blocked by the BBB that protects the brain against pathogens, but LX-2FR cells can pass BBB in this study. How to explain?

5.       In text, Oncolytic virotherapy (OV) has developed as a promising approach for GBM treatment. A promising approach to treat GBM is oncolytic virotherapy (OVT). Oncolytic viruses (OVs) are capable of replicating in neoplastic cells. Note that some of these abbreviations are simply misused.

6.       I cannot find the information of antibodies #R0409 and #F0517 in Santa Cruz Biotechnology website. Please provide the weblinks.

The manuscript is poorly organized, and language errors/inaccuracies are frequently found, such as “We used human cells that we loaded with the oncolytic virus XVir-N-31”. The authors are recommended to seek help from the professional editing service or native speakers. In addition, the comments like “The R28 cell line is further described in28”, “the area of the tumor was measured every 100 μm using ImageJ as described in10” should be revised to complete sentences, not in No of reference.

Author Response

  1. To load the LX-2FR cells with XVir-N-31, cells were infected with a multiplicity of infection (MOI) of 200 for 5 h, then directly used for INA. The infection efficiency of the viruses has not been identified.

Working with LX-2 cells for a long time now we did this experiment far before generating the fast running cells and using an adenovirus similar to XVir-N-31 which expresses GFP. Due to low expression of adenovirus receptor CAR on LX-2 cells at least 100 MOI were required to infect 90% of the cells. However, we used 200 MOI since we observed that this MOI leads to a high release of virus progeny 72 h after transduction but did not influence the cell´s motility. Parts of these data were published in (El-Ayoubi, A et al. bioRxiv 2023, 2023.08.16.553513). We have now included information about LX-2 infectability in the material and methods part of the revised manuscript.

  1. In Figure 1B, the consistency of the most obvious positive bright spots in day 3 among LN229GFP, LX-2FR, and Hexon have a great doubt. Please provide more evidence.

We apologize but we did not see the problem with the detection of the different cells as well as of the adenoviral hexon protein at day 3 post INA. Fluorescence detection channels between the three different fluorochromes we used for staining or for detection of labeled cells (GFP, mCherry) do not overlap and can be clearly separated. Therefore, mCherry positive LX-2FR cells can be clearly separated form GFP positive LN-229 glioma cells. Hexon staining patterns (as identified by a far-red-labeled secondary antibody and by this clearly separated from mCherry and GFP channels) at day 3 after INA are nearly equal to that of LX-2FR cells, indicating that XVir-N-31 replication is ongoing in these cells whilst GBM cells are clearly hexon negative. Our conclusion is substantiated by the observation that mCherry positive shuttle cells were not visible at later time points whilst at these time points hexon staining is visible in GFP positive GBM cells.

In addition, the histopathological structure of the LN229GFP transplanted tumors should be given HE staining images for further demonstration.

Histopathological structures of LN-229GFP tumors after INA of either PBS or XVir-N-31 loaded shuttle cells are given in Fig. 5B. In Fig. 5B it can be observed that in tumor harboring mice treated with INA of PBS the tumors are large and invasively spread into at least half of the hemisphere whereas LN-229GFP tumors in mice that received INA of XVir-N-31 loaded shuttle cells harbor only small tumors. We added a half-sentence regarding this issue into the results part of the manuscript. Additionally, in several other manuscripts published by our groups we demonstrated, in mice bearing human GBM xenografts that received an XVir-N-31-based oncolytic virotherapy, that this therapy induced lysis of GBM cells whilst leaving non-neoplastic brain cells and tissue unaffected (Mantwill et al, 2013; Czolk et al, 2019, Klawitter et al, 2022).

  1. In Figure 3, the second row of images, the expression of LX-2FR and HMGB1 showed a high consistency. Since LX-2FR is human cell, while HMGB1 marks a murine-derived protein, it is impossible to be in exactly the same position, so the images are not trustworthy. Similarly, in Figure 4, HMGB1 and HSP70 are labeled proteins expressed by mouse cells, while Hexon is a protein expressed by viruses. Since viruses are replicative in tumors and can spread diffusively, the viruses have a wider distribution range, and the Hexon expressed by the viruses cannot be distributed in the same position as the protein expressed by mouse tissue cells.

HBMGB1 is not exclusively a murine protein but its human analog (also named HMGB1) is expressed in human cells, therefore in GBM as well as in LX-2FR cells. We have previously demonstrated that human tumor cells infected with XVir-N-31 release HMGB1 as a sign of immunogenic cell death (Lichtenecker et al 2019; Klawitter et al, 2022), while HMGB1 is not released if tumor cells are left uninfected or are infected with either wildtype or a replication-deficient adenovirus (Klawitter et al, 2022). In this regard, in our mouse xenograft GBM model HMGB1 is not released from murine non-neoplastic cells. HMGB1 was first visible in XVir-N-31 infected LX-2FR cells at day 3 post-INA (prior to virus-mediated cell lysis) as shown by colocalization of these proteins. Later on, HMGB1 was visible in OV-infected, dying GBM cells. In addition, HMGB1 was not detectable if tumor-bearing mice were treated with INA of unloaded shuttle cells (Fig. 3 upper panel). The same is true for HSP70 which also serves as an indicator for immunogenic cell death induction (Klawitter et al, 2022).

Hexon is indeed an adenoviral protein and part of the adenoviral capsid. However, its expression is mainly visible in infected cells during virus production as the virus uses the cellular transcription and translation machinery for the synthesis of its components. Additionally, adenovirus subtype 5 (the origin of XVir-N-31) does not replicate efficiently in mouse cells (Duncan et al J. Gen Virol 1978). Summing up, this indicates that OV production is ongoing in human GBM cells. Besides, especially in Fig. 4, it is obvious that the staining patterns for hexon and HMGB1 or HSP70 are quite similar and are visible only in the area where tumor cells are located (Fig. 3), indicating a co-staining for both proteins in the same cells.  Thus, hexon and HMGB1 expression should be located in the same area, since the release of HMGB1 depends on efficient viral replication.

  1. As authors said, the entry of OVs into the brain is blocked by the BBB that protects the brain against pathogens, but LX-2FR cells can pass BBB in this study. How to explain?

Since its discovery in 2009 (Ref#14 in the manuscript) intranasal administration (INA) of cells has been shown to deliver successfully different types of stem cells in therapeutically relevant amounts to the brain along the olfactory and trigeminal nerve pathways, notably bypassing (not crossing) the BBB. This delivery route has been extensively described by a variety of studies as a feasible alternative route to the local (surgical) and systemic administration for different brain disorders (Refs #14, 15,19,31 in the manuscript) including brain tumors (Refs #16,19,20). Recently INA succeeded in the FIH clinical trial and has been proven safe in pediatric patients after INA of allogeneic stem cells (PMID: PMID: 35568047 DOI: 10.1016/S1474-4422(22)00117-X). To this end, our study provides the first direct evidence and translational background for the establishment of intranasal OV-loaded somatic differentiated cells (such as LX-2FR) for targeting invaded GBM cells alone or in combination with the local OV therapy of the tumor core.

Successful delivery of intranasally applied LX-2 cells to the brain of GBM mice has been previously proven by us, as outlined in the manuscript (Ref # 27, El-Ayoubi, A et al. bioRxiv 2023, 2023.08.16.553513).

  1. In text, Oncolytic virotherapy (OV) has developed as a promising approach for GBM treatment. A promising approach to treat GBM is oncolytic virotherapy (OVT). Oncolytic viruses (OVs) are capable of replicating in neoplastic cells. Note that some of these abbreviations are simply misused.

We apologize and corrected the abbreviations in the manuscript text.

  1. I cannot find the information of antibodies #R0409 and #F0517 in Santa Cruz Biotechnology website. Please provide the weblinks.

We apologize. Accidentally the lot number was recorded instead of the catalogue number. We thank the reviewer for pointing that out. The source of both antibodies has been corrected in the text accordingly.

  • Hexon antibody: Adenovirus hexon protein (8C4); sc-51748 https://www.scbt.com/p/adenovirus-hexon-protein-antibody-8c4?productCanUrl=adenovirus-hexon-protein-antibody-8c4&_requestid=9179277
  • YB-1 Antibody: (59-Q); sc-101198; https://www.scbt.com/p/yb-1-antibody-59-q

Comments on the Quality of English Language

The manuscript is poorly organized, and language errors/inaccuracies are frequently found, such as “We used human cells that we loaded with the oncolytic virus XVir-N-31”. The authors are recommended to seek help from the professional editing service or native speakers. In addition, the comments like “The R28 cell line is further described in28”, “the area of the tumor was measured every 100 μm using ImageJ as described in10” should be revised to complete sentences, not in No of reference.

We apologize. We rewrote parts of the manuscript and corrected mistakes. A proofread by a native speaker was performed. We hope that now the revised version of our manuscript is acceptable for publication.

Reviewer 2 Report

The manuscript deals with the topic of intranasal delivery of the oncolytic adenovirus (OAV) XVir-N-31 via virus-loaded, optimized hepatic stellate shuttle cells, LX-2FR . In particular, two orthotopic xenograft GBM mouse models were used to verify that INA of LX-2/XVir is an effective method to deliver XVir-N-31 to (invaded) GBM cells. In both these two models, oncolytic adenovirus, loaded in shuttle cells delivered by INA, replicate in and eliminate GBM cells and provide a survival benefit. In terms of mechanism, the occurrence and persistence of immunogenic cell death due to intranasal delivery of the oncolytic adenovirus xvir-n-31 via optimized shuttle cells was demonstrated.

 General concept comments:

  1. All the results presented in this article are from in vitro experiments. It would be even better if there were corresponding results from in vivo experiments to supplement them.
  2. Do infected LX-2FR express adenoviral antigens as a result of replication ? If so, does this impact the feature of infected LX-2FR? Does viral infection change the immune phenotype on LX-2FR? What advantages does this LX-2FR have to be used as a carrier to deliver the oncolytic adenovirus?
  3. The combination of IT and INA further prolonged the animal´s survival in R28GFP models, are there relative data of combination therapy in LN-229GFP models?
  4. Cytokines play an important role in the tumor immune microenvironment. Are there any available data regarding the cytokines in tumor area or in peripheral blood after treatment?
  5. In Resutls3.4, XVir-N-31 induces ICD in R28GFP derived GBMs long time after different OVT type at different timepoint, I wonder if there are differences in the ability of different OVT type to induce ICD at the same time point?

Specific comments:

  1. The sentence “In contrast, glioma stem cells (GSCs) cultured as neurospheres in stem cell medium more closely reflect the phenotype and genotype of primary tumors than do serum-cultured cell lines.” in Results3.3  have some confusion. What is the argument of this sentence? Please indicate the supporting references.
  2. The results of IHC related to apoptosis and proliferation in tumor tissues maybe further enrich this article. 

Author Response

1. All the results presented in this article are from in vitro experiments. It would be even better if there were corresponding results from in vivo experiments to supplement them.

 This comment is not comprehensible, since all experiments shown in this manuscript have been performed in vivo using two different xenograft orthotopic mouse GBM models.

2. Do infected LX-2FR express adenoviral antigens as a result of replication? If so, does this impact the feature of infected LX-2FR? Does viral infection change the immune phenotype on LX-2FR? What advantages does this LX-2FR have to be used as a carrier to deliver the oncolytic adenovirus?

Since viral replication will produce viral proteins, infected LX-2FR cells surely will produce adenoviral antigens. Thus, viral infection will certainly change the immune phenotype of these cells. However, we did not analyze this issue in detail because, this might not influence the therapeutic impact of the LX-2/XVir-based OVT, since OV-loaded shuttle cells will be rapidly eliminated due to virus replication and subsequent virus-mediated cell lysis as demonstrated in Fig. 1 and 3.

The advantage of delivering OVs by INA of virus-loaded shuttle cells is the application route since a direct injection into the tumor is invasive and requires a complex surgical intervention bearing risks for the patient. As mentioned several times in the manuscript, notably in the introduction and discussion parts, INA of OV-loaded shuttle cells also hits those infiltratively growing GBM cells that cannot be eliminated if the OV (without shuttle cells) is applied intratumorally due to the “shielding” provided by non-neoplastic cells to those brain-invaded GBM cells that are located far away from the core tumor. These cells are surrounded by non-neoplastic cells which restricts virus replication and therefore virus spreading. Thus, the chance that virus progeny generated in the core tumor will reach invaded GBM cells is extremely low.

3. The combination of IT and INA further prolonged the animal´s survival in R28GFPmodels, are there relative data of combination therapy in LN-229GFP models?

Due to the Protection of Animals Act we have to use as few animals in experiments as possible. Therefore, we performed the combination therapy only in R28GFP GBM mice since R28 GBMs (compared to LN-229 GBMs) present highly infiltratively growing tumors and cannot be cured by an intratumoral injection of XVir-N-31 (Mantwill et al, 2013). Additionally, the R28 GBM orthotopic mouse model reflects a more clinically relevant mouse GBM than the LN-229 model. R28 is a glioma stem cell-derived cell line and despite being less established, it resembles more of the features of human primary GBM cells. We added a reference in the text part to address this point. In contrast to LN-229 cells, R28 cells are grown as neurospheres without FCS, but in the presence of bFGF, EGF and LIF which protects cell differentiation. Additionally, and in contrast to LN-229 cells, R28 cells are temozolomide resistant as most human GBMs are. 

4. Cytokines play an important role in the tumor immune microenvironment. Are there any available data regarding the cytokines in tumor area or in peripheral blood after treatment?

We have not analyzed this topic in this setting, because we were focusing on the delivery of XVir-N-31 to GBMs in vivo and the therapeutic impact of our INA approach. However, in the publication of Schober et al (Clinical Cancer Research, 2023; PMID: 36892582) we described this issue in detail showing that CXCL10 plays an important role in the innate and adaptive anti-tumoral immune response induced by an XVir-N-31 based OVT.

5. In Results 3.4, XVir-N-31 induces ICD in R28GFPderived GBMs long time after different OVT type at different timepoint, I wonder if there are differences in the ability of different OVT type to induce ICD at the same time point?

This is an important point.  We analyzed ICD induction in the R28 GBM model only at the endpoint at the time the animals had to be sacrificed due to the development of tumor-associated symptoms. However, for all analyzed human GBM cells we tested so far, we observed an induction of ICD (measured by the release for DAMPs such as HMGB1 or HSP70 or by the expression of the “eat me” protein calreticulin on the surface of XVir-N-31 infected GBM cells) much earlier (demonstrated in this manuscript for LN-229 tumors and in our previous publication (Klawitter et al, IJMS, 2022)). In the R28 model our main interest was whether ICD can be detected even a long time after the treatment. Surprisingly, both XVir-N-31 replication and ICD induction were visible even months after OVT. As ICD is strictly dependent on OV replication, the different types of OVT might influence ICD. However, modern translational research on OVT lacks currently available methods of ICD quantification to determine differences in its induction by different application techniques of the OV. The authors believe that the most efficient OVT that also hits invaded tumor cells might also induce an optimal induction of ICD. In this regard we have observed that a second boost of ICD in GBMs can be induced not only by virus replication, but also by activated, tumor-targeting and tumor-infiltrating T cells (Klawitter et al, 2022).

ICD also depends on the species of OV that is used for OVT (Ma et al, Cell Death Diss 2020). In this regard we have demonstrated that XVir-N-31 is a potent inducer of ICD (Klawitter et al, 2022; Lichtenecker et al, 2019). In contrast, adenovirus wild type expressing the large E1A13S protein, or replication deficient adenovirus, do not induce an immunogenic phenotype of cell death. Consequently, OVs based on expression of E1A13S show very likely a reduced capacity of ICD induction. This issue has been discussed in Klawitter et al (2022).

Specific comments:

  1. The sentence “In contrast, glioma stem cells (GSCs) cultured as neurospheres in stem cell medium more closely reflect the phenotype and genotype of primary tumors than do serum-cultured cell lines.” in Results 3.3 have some confusion. What is the argument of this sentence? Please indicate the supporting references.

We thank the reviewer for this comment and have added some information about this issue in the results part of the manuscript.

 2. The results of IHC related to apoptosis and proliferation in tumor tissues maybe further enrich this article. 

 This issue was not the focus of the recent study. The study mainly aimed at the investigation of XVir-N-31 delivery to the GBM by INA of a new type of shuttle cells and at the assessment of the therapeutic impact of this application method. It is long known and widely accepted that OVs kill tumor cells by oncolysis and, as a secondary effect, by the induction of ICD and therefore induction of a tumor-specific immune response (Klawitter et al, 2022). Concerning the oncolytic potency of XVir-N-31 (also named Ad-Delo3RGD), more than 50 publications, co-authored by us, have determined the therapeutic effects of this OV in several tumor entities. It has been also recently shown by our collaborators that an intervention of the cell cycle influences virus replication (Ehrenfeld et al, 2023; PMID: 37219458) and by this the impact of an XVir-N-31 based OVT.

Reviewer 3 Report

Reviewing Cancers2595623

Attention of authors is drawn towards the following points

1.       The mouse model used in this study may be differently responding to a treatment as compared to mice endogenously developing GBM.

2.       A multiplicity of infection of 200 used for preparing the OV infected cells is too high unless an optimization of the protocol is done and found suitable for the purpose, High MOIs can cause lysis from without.

3.       Images presented in Figure 1 show disappearance/lysis of the shuttle cells 12- and 18-days post INA while the virus colocalizes in tumor cells. Tumor cells are not lysed till 18 days post INA.

4.       In legend to Figure 3, the term “virus-unloaded cells” should be replaced by “cells without virus” or simply “unloaded cells”.

5.       In Figure 3 also, the appearance of the outline of 18 days post-INA would indicate intactness/survival of the cells/ Expression of markers of cell death have to result in withering and disappearance of cells, if I am not mistaken.

6.       It is not clear why weight of control animas was not monitored till the end of the experiment as in Figure 5. Also, the days mentioned on the x-axis are 1,16, 31, 42, 50, and 64 while the graph shows values corresponding to every third or fourth day. It is a strange set of numbers chosen for some reason which is not clear.

7.       The lifespan of mice as in figures 5 and 6 are not comparable even for controls

8.       The persistence of virions/hexon proteins (Figure 7) should be from within infected cells. However, it is desirable that it phases out with time. Are we sure that they would not continue inducing lysis of non-cancerous cells in this situation? It is good to see hexon expression in the infiltration zone.

9.       The authors must comment on the distribution pattern of HMGB1 fluorescence which is localized in IT and INA but is more diffuse in Combi.

10.   There are no major problems with the organization of the manuscript or with the English language except for a few typographical errors highlighted in the pdf being reloaded.

Author Response

  1. The mouse model used in this study may be differently responding to a treatment as compared to mice endogenously developing GBM.

The mouse models we used are well-established animal models of GBM (also for OVT approaches) and are described in many publications including ours (e.g. Mantwill et al, 2013; Czolk et al, 2019; Schötterl et al, 2019; Ranjan et al, 2021; Klawitter et al, 2022). Since adenovirus subtype 5 (the origin of XVir-N-31) does not replicate in mouse cells it cannot be tested in mice developing endogenous tumors as these will be of murine origin.

Compared to human GBM cells, mouse GBM cells might respond differently to a treatment. Since the primary aim is to treat human tumors in our opinion a xenograft model should be prioritized over the murine models of GBM to demonstrate the therapeutic impact of the novel treatment approach described in our manuscript and intended to be translated into human use. In this regard we added an explanation sentence to the results part, chapter 3.1.

Unfortunately, in contrast to endogenous mouse GBM models that are fully immune-competent, the induction of an anti-tumoral immune response, based on the OV-mediated induction of ICD, could not been demonstrated in the xenograft model we used in this study as immune-deficient mice are mandatory for the development of human tumors. However, the induction of an anti-tumoral immune response by XVir-N-31 has been demonstrated in immune-humanized mice in a recent publication of our group (Klawitter et al, IJMS, 2022). This issue was mentioned in the discussion part of the manuscript.

 2. A multiplicity of infection of 200 used for preparing the OV infected cells is too high unless an optimization of the protocol is done and found suitable for the purpose, High MOIs can cause lysis from without.

Working with LX-2 cells for a long time, in the past we performed control experiments regarding this issue using an adenovirus expressing GFP. Due to low expression of the adenovirus receptor CAR on LX-2 cells at least 100 MOI are necessary to infect 90% of the cells. However, we used 200 MOI since we observed that this MOI leads to a high release of virus progeny 72 h after transduction but did not influence cell motility after infection. Besides, XVir-N-31 replication in LX-2 cells is delayed compared to human GBM cells and commencing lysis at an infection rate of 200 MOI was observed earliest 60 h post-infection. Parts of these data were published in (El-Ayoubi, A et al. bioRxiv 2023, 2023.08.16.553513, Ref#27 in the manuscript). Additionally, as shown in Fig. 1B, LX-2FR cells loaded with 200 MOI of XVir-N-31 were detected in the tumor area 72 h after infection, indicating that the cells reached the tumor area.

 3. Images presented in Figure 1 show disappearance/lysis of the shuttle cells 12- and 18-days post INA while the virus colocalizes in tumor cells. Tumor cells are not lysed till 18 days post INA.

There seems to be a misinterpretation of our data. Figure 1 indicates that after the virus-mediated lysis of LX-2FR shuttle cells the virus spreads to and replicates in the GBM cells, showing that virus replication is ongoing in the tumor now. However, LN-229 cells are fast proliferating GBM cells with a duplication time of approximately 24 h. As the XVir-N-31 replication cycle in LN-229 cells lasts appr. 48 h, at an early time point after INA the oncolytic effect induced by XVir-N-31 might be slower than the proliferation rate of the tumor cells. In addition, lysed GBM cells cannot be detected here as they lose GFP positivity. At later time points after INA, we believe that there is a chain reaction of virus production, and tumor cell lysis will increase over time as shown by the virus effect on the animal survival (Fig. 5). 

 4. In legend to Figure 3, the term “virus-unloaded cells” should be replaced by “cells without virus” or simply “unloaded cells”.

The legend of Figure 3 is corrected accordingly.

5. In Figure 3 also, the appearance of the outline of 18 days post-INA would indicate intactness/survival of the cells/ Expression of markers of cell death have to result in withering and disappearance of cells, if I am not mistaken.

This is a good argument! Nevertheless, HMGB1 and HSP70 are not direct markers of cell death, but rather “Damage Associated Molecular Patterns” (DAMPs) proteins that will be released by cells in which ICD is induced. The protein level of HMGB1 seemed to be elevated in XVir-N-31 infected GBM cells and will be released from these cells even before they die and disappear (published in our recent paper (Klawitter et al, 2022). As mentioned in our answer to comment #3, LN-229 cells are fast proliferating GBM cells with a duplication time of approximately 24 h. As the XVir-N-31 replication cycle in LN-229 cells lasts approximately 48 h, at earlier time points after INA the oncolytic effect induced by XVir-N-31 might be slower than the proliferation rate of the tumor cells. Additionally, lysed LN-229 GBM cells cannot be detected here as they lose GFP as well as HMGB1 positivity, whereas OV-infected cells that have not been lysed so far show both, GFP and HMGB1 expression, and cells that were not infected 18 days after INA are only GFP positive.

In this regard we have modified Fig. 3 by indicating hot spots of ICD induction as well as areas of more distributed ICD signs in the tumor area. We also added text description regarding this change in the results and discussion part of the manuscript.

6. It is not clear why weight of control animals was not monitored till the end of the experiment as in Figure 5. Also, the days mentioned on the x-axis are 1,16, 31, 42, 50, and 64 while the graph shows values corresponding to every third or fourth day. It is a strange set of numbers chosen for some reason which is not clear.

Control mice were monitored until they had to be sacrificed due to tumor associated symptoms. As the last control mice had to be sacrificed at day 41 (INA:PBS) and day 42 (INA:LX2: refer to Fig. 5C), monitoring of their weight at later time points was not possible. The ticks of the X-axis in Fig. 5E and 6E were randomly generated by GraphPadPrism and are reformatted now to display regularly spaced ticks.

 7. The lifespan of mice as in figures 5 and 6 are not comparable even for controls

This is true, Fig. 5 and 6 show survival analyses for two different mouse GBM models with completely different dynamics of tumor growth. Fig. 5 presents data from mice bearing human LN-229 tumors. These tumors grow fast as the doubling time of LN-229 cells is around 24 h, and the median survival of mice is approximately 38 days. This survival time for the LN-229 GBM bearing mice is common and repeatedly reported in a variety of studies by us and others. In contrast, R28 cells are slowly proliferating glioblastoma derived stem cells, and mice harboring orthotopic R28 derived GBMs live significantly longer (Mantwill et al, 2013). R28 tumors, in contrast to LN-229 GBMs, show massive infiltration into the healthy brain parenchyma. We therefore used the R28 GBM in addition to the LN-229 mouse model to demonstrate that INA of XVir-N-31 loaded LX-2FR cells can kill invaded GBM cells. The differences between the two mouse models have been demonstrated and discussed in the manuscript.

  1. The persistence of virions/hexon proteins (Figure 7) should be from within infected cells. However, it is desirable that it phases out with time. Are we sure that they would not continue inducing lysis of non-cancerous cells in this situation?

XVir-N-31 replication is dependent on nuclear YB-1, a protein which is not expressed in normal brain tissue (Mantwill et al, 2013). Additionally, no replication of XVir-N-31 was detected in human astrocytes (Mantwill et al, 2013). No XVir-N-31 replication was also observed in any tissue of Syrian Hamsters that are permissive for adenovirus replication and that have been used in toxicity studies to prove virus safety regarding its clinical use. Therefore, it is very unlikely that XVir-N-31 will replicate in non-neoplastic cells. Furthermore, additional safety and toxicity studies have been performed according to regulatory requirements of clinical trials in Germany (Regulation of admission) as the clinical trial for the use of XVir-N-31 to treat recurrent GBM patients is anticipated to start in 2024. Besides that, XVir-N-31 does not replicate in mouse cells, therefore mouse brain tissue will be infected, but no virus replication will occur. Subjected to these prerequisites, the phasing out with time could be also a result of a bulk of virus particles located in the tissue surrounding infected cells which can be detected with the hexon antibody we used as hexon is a protein of the virus capsid.

Regarding the distribution of HMGB1 in IT, INA and mice with combined treatment we added some sentences in the results part and discussed this issue in more detail in the discussion part of the revised manuscript.

 It is good to see hexon expression in the infiltration zone.

This is a very important point in the clinical setting. This is certainly of advantage, since in GBMs the infiltration zone is very likely the source of tumor relapse.

 9. The authors must comment on the distribution pattern of HMGB1 fluorescence which is localized in IT and INA but is more diffuse in Combi.

Indeed, in the photographs we show in Fig. 8 differences in the distribution of HMGB1 fluorescence in the different treatment groups are visible. We now analyzed the other mice of each treatment group and did not observe significant differences in the distribution of HMGB1 in INA, IT and Combi treated mice. However, in some tumor areas, hot spots of HMGB1 staining were observed whereas in others the staining was more diffused and in some even absent. As ICD induction (identified by HMGB1 staining) is strictly dependent on XVir-N-31 replication we believe that the hot spots of HMGB1 staining are also hot spots of OV replication. Images from two additionally analyzed mice in each group are added to Fig. 8 as well as the respective text description is provided in the results and discussion part of the manuscript.

 10. There are no major problems with the organization of the manuscript or with the English language except for a few typographical errors highlighted in the pdf being reloaded.

We thank the reviewer for this comment and corrected the typos in the text.

Round 2

Reviewer 1 Report

After the author's revision, some problems were resolved and no further comments.

Author Response

.
